# Use of trifluoroacetaldehyde N-tfsylhydrazone as a trifluorodiazoethane surrogate and its synthetic applications

Xinyu Zhang[1], Zhaohong Liu[1], Xiangyu Yang[1], Yuanqing Dong[1], Matteo Virelli[2], Giuseppe Zanoni[2], Edward A. Anderson [3] & Xihe Bi[1,4]

Trifluorodiazoethane ($CF_3CHN_2$), a highly reactive fluoroalkylating reagent, offers a useful means to introduce trifluoromethyl groups into organic molecules. At present, $CF_3CHN_2$ can only be generated by oxidation of trifluoroethylamine hydrochloride under acidic conditions; due to its toxic and explosive nature, its safe generation and use remains a prominent concern, hampering wider synthetic exploitation. Here we report the development of tri-fluoroacetaldehyde N-tfsylhydrazone (TFHZ-Tfs) as a $CF_3CHN_2$ surrogate, which is capable of generating $CF_3CHN_2$ in situ under basic conditions. The reaction conditions employed in this chemistry enabled a difluoroalkenylation of X–H bonds (X = N, O, S, Se), affording a wide range of heteroatom-substituted *gem*-difluoroalkenes, along with Doyle-Kirmse rearrangements and trifluoromethylcyclopropanation reactions, with superior outcomes to approaches using pre-formed $CF_3CHN_2$. Given the importance of generally applicable fluorination methodologies, the use of TFHZ-Tfs thus creates opportunities across organic and medicinal chemistry, by enabling the wider exploration of the reactivity of trifluorodiazoethane.

[1] Department of Chemistry, Northeast Normal University, 130024 Changchun, China. [2] Department of Chemistry, University of Pavia, Viale Taramelli 12, 27100 Pavia, Italy. [3] Chemistry Research Laboratory, University of Oxford, 12 Mansfield Road, Oxford OX1 3TA, UK. [4] State Key Laboratory of Elemento-Organic Chemistry, Nankai University, 300071 Tianjin, China. These authors contributed equally: Xinyu Zhang, Zhaohong Liu. Correspondence and requests for materials should be addressed to X.B. (email: bixh507@nenu.edu.cn)

Trifluorodiazoethane (CF$_3$CHN$_2$, also known as tri-fluoromethyldiazomethane) is a highly reactive tri-fluoromethylating agent employed in transformations such as cycloadditions[1–6], X–H insertions[7,8], coupling reactions[9,10] and homologations (Fig. 1a)[11,12]. CF$_3$CHN$_2$ is generated by the oxidation of trifluoroethylamine hydrochloride (CF$_3$CH$_2$NH$_2$·HCl) under acidic conditions, but being a toxic and explosive gas, handling of CF$_3$CHN$_2$ at room temperature is extremely hazardous if a significant buildup occurs[2,13]. Although first described in 1943[14], only in the last decade have improve-ments to this method been made, involving slow addition of aqueous NaNO$_2$ to trifluoroethylamine to avoid an accumulation of large amounts of CF$_3$CHN$_2$[2]. More recently, other operational improvements have been developed, such as the small-scale preparation of CF$_3$CHN$_2$ in solution[5,7,8], the recycling of gaseous CF$_3$CHN$_2$[9], and the use of continuous-flow chemistry[15–17]. Nevertheless, the use of specifically designed equipment and/or operating conditions remains a general limitation, and reagents that are easily handled but capable of slowly generating CF$_3$CHN$_2$ in situ under mild conditions (ideally complementing current oxidative/acidic methods) are of high appeal.

We targeted the use of trifluorinated N-sulfonylhydrazones as a trifluorodiazoethane surrogate. As a class of stable precursors to diazo compounds, sulfonylhydrazones are widely used in synth-esis[18–22]; however, trifluoroacetaldehyde-derived sulfonylhy-drazones have not been explored as surrogates for CF$_3$CHN$_2$[23,24]. gem-Difluoroalkenes are important motifs in the design of mechanism-based enzyme inhibitors, and as bioisosteres of the carbonyl group[25-27], and are typically prepared by Wittig or Julia–Kocienski-type reactions[28–30], or by cross-coupling[31–34]. However, these methods are mostly effective only for the synth-esis of C-substituted gem-difluoroalkenes, while the synthesis of their heteroatom-substituted counterparts is comparatively rare and suffers from narrow substrate scope, or requires strong bases or toxic reagents[35–39].

We report here the development of trifluoroacetaldehyde N-tfsylhydrazone (TFHZ-Tfs) as a bench-stable CF$_3$CHN$_2$ pre-cursor, which decomposes in a controlled manner under basic conditions to release CF$_3$CHN$_2$ into the reaction system (Fig. 1b); this strategy circumvents the need for slow addition or manual handling of CF$_3$CHN$_2$, thus minimizing exposure and reducing the potential explosion risk. Importantly, this base-mediated approach also led to the discovery of novel reactivity of CF$_3$CHN$_2$: we describe its use in the difluoroalkenylation of X–H (X = N, O, S, Se) bonds, overcoming limitations in previous routes to these motifs, and also in Doyle–Kirmse and cyclopro-panation reactions, which display excellent stereoselectivity and yields, and collectively demonstrate the potential utility of TFHZ-Tfs as a trifluorodiazoethane surrogate.

## Results

**Synthesis of TFHZ-Tfs.** TFHZ-Tfs could be easily accessed by condensation of the o-trifluoromethylbenzenesulfonyl hydrazide with trifluoroacetaldehyde monohydrate under acidic conditions. The reaction proved readily scalable, TFHZ-Tfs could be pre-pared in high yield (91%) on 85 mmol scale as a bench-stable crystalline solid, and in a cost-effective manner, which is attrac-tive for synthetic applications. In addition, TFHZ-Tfs could be stored at ambient temperature for at least 5 months without degradation (as characterized by $^1$H NMR spectroscopy).

**Investigation of reaction conditions** . An exploration of the reactivity of TFHZ-Tfs began in the difluoroalkenylation of X–H bonds. p-Methylthiophenol was identified as a suitable nucleo-phile for this study, and to our delight we found that in the

presence of aqueous KOH, sodium dodecylbenzenesulfonate (SDBS, 30 mol%), and the iron porphyrin catalyst FeTPPCl (5 mol%) in dichloromethane at 40 °C, TFHZ-Tfs delivered the difluoroalkenylated product **2** in 51% yield, along with 9% of the trifluoroethyl thioether **2′** (Fig. 2, Entry 1). Iron porphyrin complexes have been applied as highly efficient catalyst in carbene-transfer reactions[40]. Screening of other iron porphyrin complexes led to the discovery of the more robust Fe[P2] catalyst, which at just 1 mol% loading afforded **2** in 80% isolated yield, while suppressing the formation of side product **2′** (Entries 2 and 3). Under the same conditions, TFHZ-Ns and TFHZ-Ts gave **2** in significantly lower yield (Entries 4 and 5). Additional optimiza-tion of this S–H gem-difluoroalkenylation led to refinement of the reactions parameters (Entry 3, TFHZ-Tfs (2.0 equiv), 5 mL KOH aq. (20 wt%), and SDBS (30 mol%) in the presence of 1 mol% of Fe[P2] in DCM at 40 °C under air; see Supplementary Table 1 for details).

**Scope of thiol gem-difluoroalkenylation.** Having established the decomposition profile of TFHZ-Tfs, and conditions for thiol difluoroalkenylation, the scope of this insertion was explored. Under the optimized conditions of Fig. 2 Entry 3 (Method A), a broad tolerance of arene substituents was observed (Fig. 3), with thiophenols bearing both electron-donating and electron-withdrawing substituents giving the corresponding difluoroalk-enes in good to excellent yields (**2–19**). Notably, reaction effi-ciency was not compromised by the positioning of the aryl substituent (ortho, meta, or para), and indeed sterically hindered mono- or bis-ortho-substituted substrates afforded the difluor-oalkenes in high yields (**20–21**). Thienyl, furyl, and 2-naphthalene thiols were also excellent substrates, leading to heteroaryl- and naphthyl sulfides **22–24**. We were pleased to find that benzene-selenol performed equally well, affording the selenodifluoroalkene **25** in 66% yield. The difluoroalkenylated structure was unam-biguously confirmed by single crystal X-ray diffraction analysis of sulfone **9′**, which was prepared by oxidation of **9** with m-CPBA (see Supplementary Table 7 for X-ray crystallographic data).

**Scope of amine gem-difluoroalkenylation.** We next questioned whether other heteroatoms could also serve as suitable nucleo-philic coupling partners, and turned our attention to amine difluoroalkenylation. After extensive screening of reaction para-meters, a copper catalyst system was identified that efficiently mediated this transformation, consisting of Cu(OTf)$_2$ (20 mol%) and LiO$t$-Bu (4.0 equiv) in DCE:toluene (3:1) under argon at 40 °C (Fig. 3, Method B, see Supplementary Table 2 for details of reaction optimization). The reaction scope encompassed a variety of aniline derivatives, with ring substituents including halides, nitriles, ketones, esters, and anthraquinones, delivering the gem-difluoroenamines in moderate to good yields (**26–35**). In some cases, incomplete conversions were observed, but the residual amine substrate could be recovered. In addition to primary ani-lines, benzophenone imine proved a viable substrate: product **36** was obtained in 51% yield, suggesting this method could be applied to the N-difluoroalkenylation of other nitrogen-based nucleophiles. Secondary amines did not prove suitable, as shown by the low yield of compound **37**.

**Scope of alcohol gem-difluoroalkenylation.** Further expansion of the scope of the methodology was achieved through mod-ification of the copper catalyst to enable the synthesis of difluorovinyl ethers from alcohols (Fig. 3, Method C, for details of optimization see Supplementary Table 3). A wide selection of benzyl-, alkyl-, and heteroaryl-substituted alcohols afforded gem-difluorovinyl ethers in good to excellent yields; for benzyl

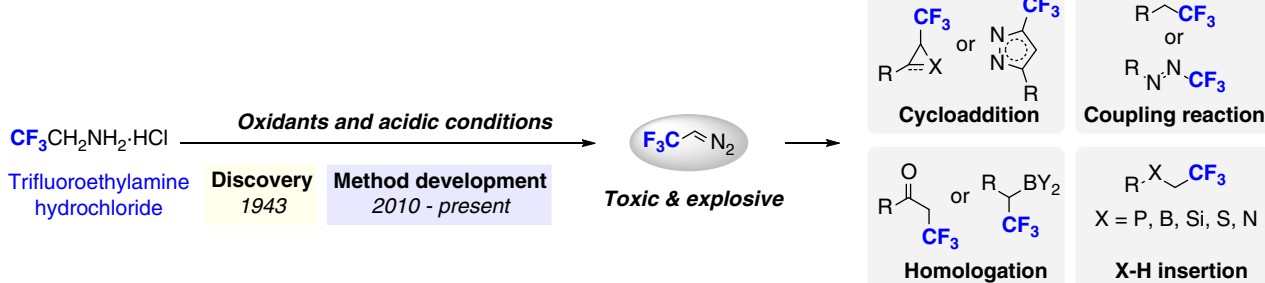

**Fig. 1** Generation and transformations of trifluorodiazoethane, and synthesis of TFHZ-Tfs. **a** Synthesis and applications of trifluoromethyldiazomethane (CF$_3$CHN$_2$) in organic synthesis. CF$_3$CHN$_2$ is a highly reactive trifluoroalkylation reagent, but its simplex synthesis method, inherent toxicity and explosiveness limit its widespread application. Because of its hazardous nature, manifold methods have been developed for the safer use of CF$_3$CHN$_2$ such as slow addition of oxidants, small-scale preparation of CF$_3$CHN$_2$ solution, recycling of gaseous CF$_3$CHN$_2$ and continuous-flow chemistry. **b** Method for the generation of CF$_3$CHN$_2$ from trifluoroacetaldehyde N-tfsylhydrazone under basic condition and *gem*-difluoroalkenylation of X–H

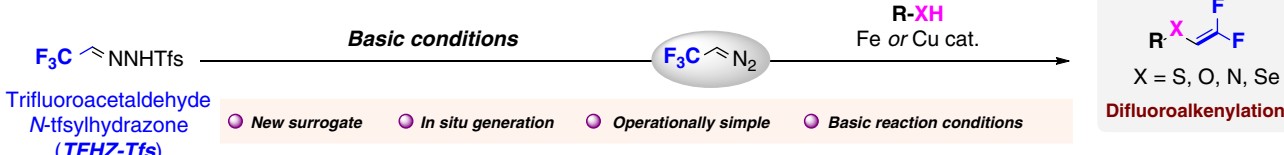

**Fig. 2** Optimization of the iron-catalyzed *gem*-difluoroalkenylation of *p*-methylthiophenol with trifluoromethyl sulfonylhydrazones. Reaction conditions: thiophenol (0.3 mmol), sulfonylhydrazone (0.6 mmol), Fe porphyrin catalyst, SDBS (sodium dodecylbenzenesulfonate) (0.09 mmol), DCM (1.0 mL), and KOH solution (5.0 mL, 20% wt %), 40 °C, 18 h, under air. [a]Yields determined by [1]H NMR spectroscopic analysis with CH$_2$Br$_2$ as an internal standard. [b]Reaction carried out under Ar atmosphere. [c]Yield in parentheses is the isolated yield

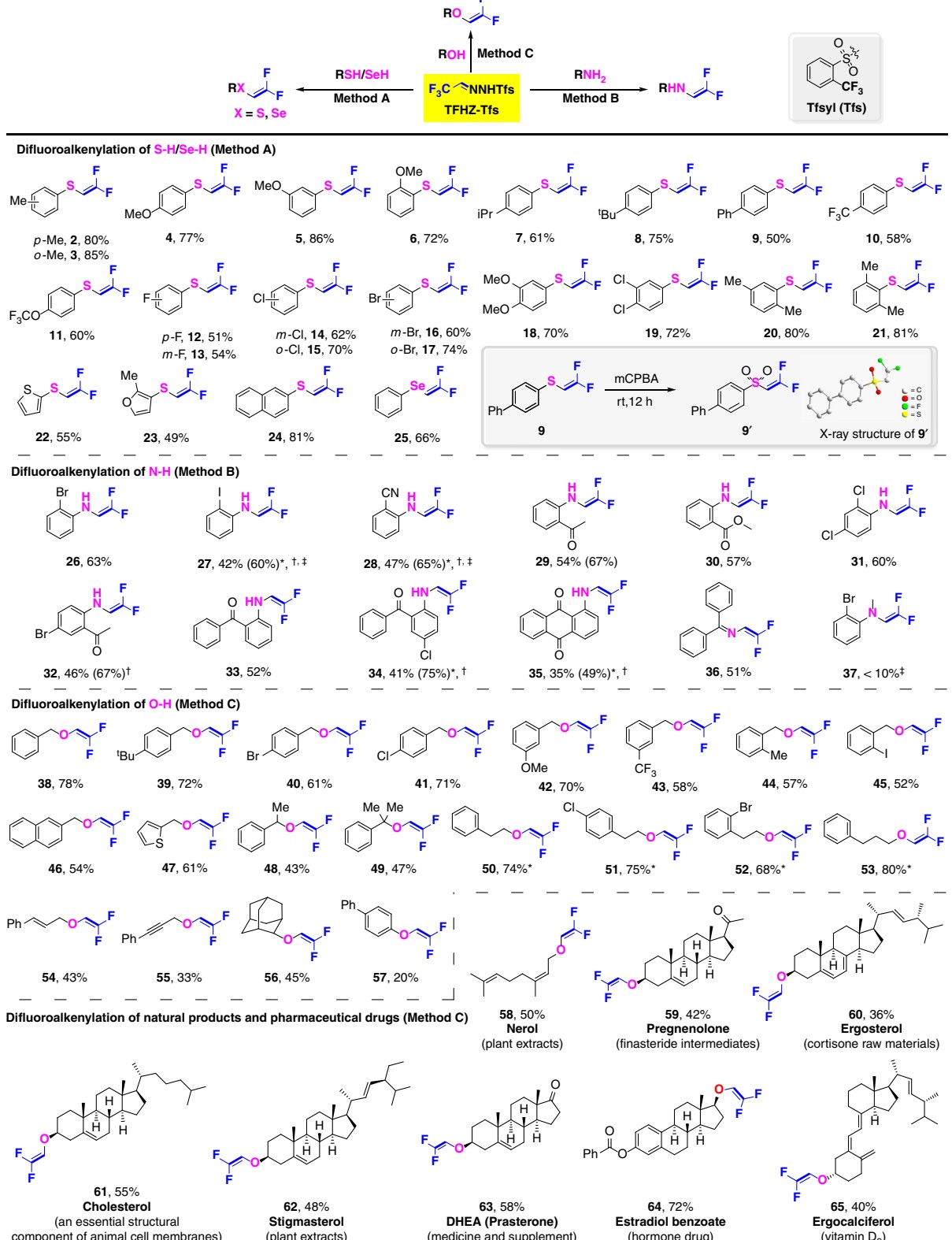

**Fig. 3** Scope of *gem*-difluoroalkenylation of X–H (X = N, O, S, Se). Reaction conditions: Method A: thiophenol (0.3 mmol), TFHZ-Tfs (0.6 mmol), Fe[P2] (1 mol%), SDBS (30 mol%), KOH (aq.)/DCM (5:1), air, 40 °C, 18 h. Method B: amine (0.3 mmol), TFHZ-Tfs (0.6 mmol), Cu(OTf)$_2$ (20 mol%), LiO$^t$Bu (4 equiv), DCE: toluene (3:1), Ar, 40 °C, 24 h. Method C: TFHZ-Tfs (1.0 mmol), NaH (4 equiv) and DCE (8.0 mL) were stirred at rt for 1 h under Ar, then CuBr (30 mol%), alcohol (0.5 mmol), and LiO$^t$Bu (1 equiv) were added and the mixture was stirred at 40 °C under Ar for 24 h. *Reaction performed for 30 h. †Number in parentheses is the yield based on recovered starting material (brsm). ‡The yield was determined by $^1$H NMR spectroscopic analysis with CH$_2$Br$_2$ as an internal standard

alcohols, the position of substituents on the arene had little influence on the reaction outcome (**38–47**), and secondary and tertiary benzyl alcohols also afforded the corresponding products with respectable efficiency (**48** and **49**). Alkyl alcohols (such as phenethyl and phenylpropyl), and other functionalized alcohols (such as cinnamyl, propargyl, and 2-adamantyl), all proved reactive partners, affording products **50–56** in moderate to high yields. In contrast to aliphatic alcohols, phenols showed poor reactivity; for example, 4-biphenylol gave the difluoroalkenylated product **57** in just 20% yield, which presumably reflects the poorer nucleophilicity of the phenol compared to the aliphatic substrate.

**Scope of *gem*-difluoroalkenylation with bioactive molecules**. To illustrate potential utility, the methodology was applied to the *gem*-difluoroalkenylation of selected natural products, drugs, and pharmaceutical intermediates. For instance, various terpene and steroid natural products (nerol, pregnenolone, ergosterol, cholesterol, and stigmasterol) were derivatized into the desired difluorovinyl ethers in good yields (**58–63**). Estradiol benzoate, a highly potent hormone therapy agent used to treat estrogen deficiencies, could also be converted to the corresponding *gem*-difluoroalkenylated product **64** in 72% yield. Further, the synthesis of *gem*-difluoroalkenylated vitamin $D_2$ **65**, (the parent being a potent drug for treatment of cutaneous tuberculosis and lupus erythematosus), was achieved in the presence of its potentially sensitive triene functionality, underlining the functional group tolerance of this methodology. It is notable that fluoroalkyl ethers represent the key structure of many insecticides and lubricants; the ready availability of such *gem*-difluorovinyl ethers may provide new opportunities for the design and construction of such molecules[41].

**Gram-scale synthesis and further transformations**. For multigram-scale applications, the Fe[P2] catalyst (which requires a multistep synthesis) could be conveniently replaced with the commercially available FeTPPCl (Fig. 4). Using this alternative catalyst with dichloromethane as solvent, *gem*-difluorovinyl sulfide **9** was obtained in a yield of 47%, which is comparable to that obtained with Fe[P2] (Method A, 50%). Interestingly, this product could be smoothly mono-defluorinated by treatment

with CuCl and $B_2pin_2$ to give the (Z)-monofluorovinyl sulfide **66** in 67% yield[42]; to our knowledge, no other routes to selectively access such monofluorinated alkenyl thioethers are known. Alternative functionalization also proved possible, such as substitution of both fluorines in **9** by reaction with excess *p*-methoxyphenol in the presence of NaH, affording the trisubstituted olefin **67** in 42% yield[43].

**Mechanistic investigations**. To gain insight into the reaction mechanism, a $CF_3CHN_2$ solution in dichloromethane was prepared according to previous reports (Fig. 5, Eq. 4)[5], and then treated with 4-methylbenzenethiol under Method A, which led to **2** and **2′** in 48% and 3% yield respectively as detected by [1]H NMR analysis of the crude reaction mixture. However, if neutral water was used instead of aqueous KOH, only **2′** was obtained (61%). These results suggest that base plays a crucial role in the *gem*-difluoroalkenylation reaction, in that it may either facilitate reaction of the heteroatomic nucleophile by deprotonation, and/or may promote a β-F elimination of a reaction intermediate. To confirm that the trifluoroethyl sulfide **2′** is indeed a side product rather than an intermediate, conversion of **2′** into **2** under Method A was attempted, but without success (Eq. 5). This may imply that the reaction mechanism involves direct fluoride ion elimination from a carbenoid-derived species, rather than elimination of HF from the trifluoroethyl group. The observation that (2,2,2-trifluoroethoxy)methylbenzene **68** was also not converted to **38** under Method C (Eq. 6) supports this hypothesis.

**Proposed mechanism**. Based on these experimental results, a plausible reaction mechanism is proposed in Fig. 6 in which trifluorodiazoethane is generated in situ from TFHZ-Tfs under the basic reaction conditions, and then reacts with the metal catalyst to form carbenoid intermediate **A**. The latter is trapped by the substrate (or deprotonated substrate) to form the oxonium ylide **B**[44]; Following deprotonation under the basic reaction conditions, the resultant intermediate **C** undergoes β-fluoride elimination to give the *gem*-difluoroalkenylation product, regenerating the metal catalyst[34,45].

**Applications of TFHZ-Tfs in Doyle–Kirmse reaction**. The high efficiency observed in the difluoroalkenylation encouraged us to

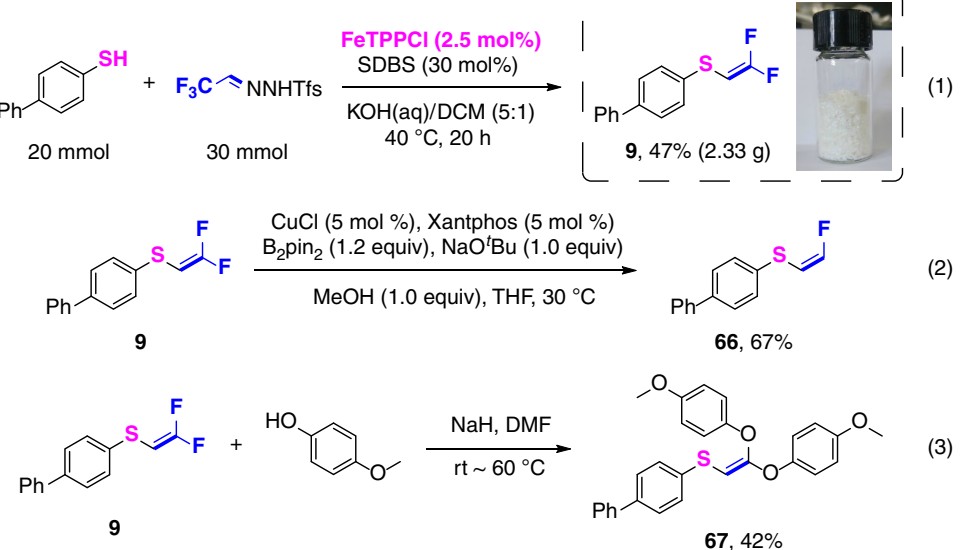

**Fig. 4** Gram-scale synthesis and further transformations. Gram-scale synthesis of product **9** (1). Mono-defluorination of product **9** (2). Double defluorination of product **9** (3)

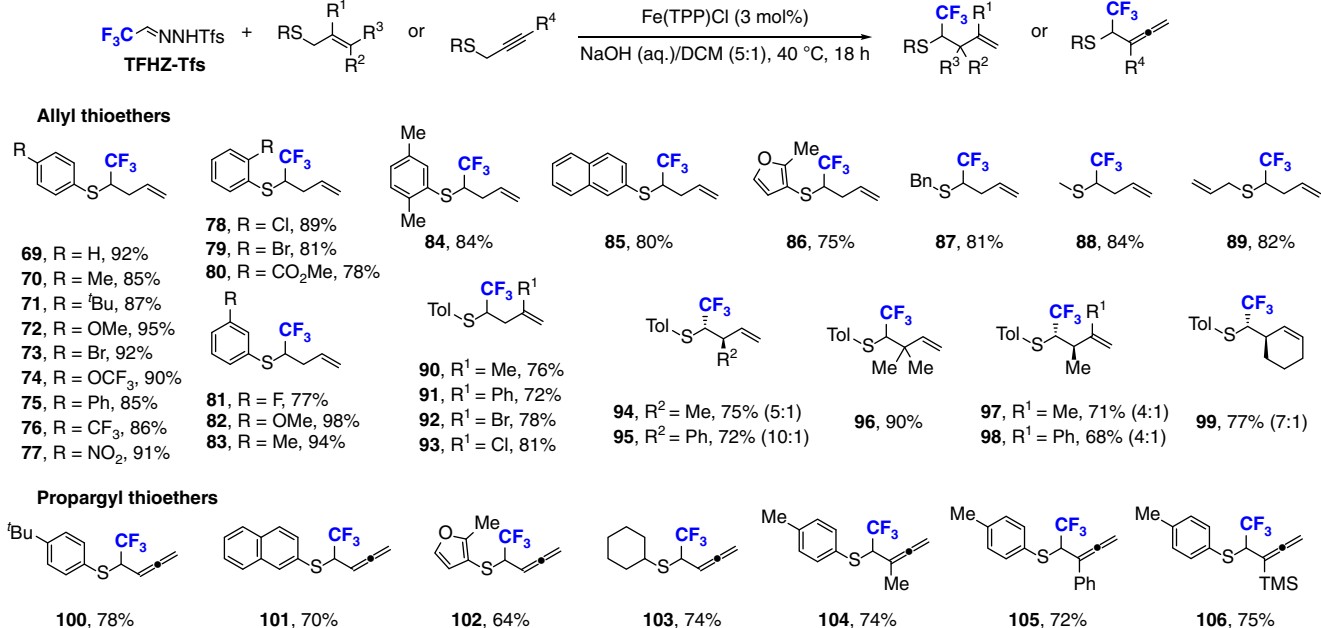

**Fig. 5** Mechanistic investigations. Base promotes fluoride elimination (4) Intermediate experience verification (5) and (6)

**Fig. 6** Proposed mechanism. Mechanistic insights regarding to formation of oxonium ylide and fluoride elimination

**Fig. 7** Scope of Doyle–Kirmse reaction. Reaction conditions: thioether (0.3 mmol), TFHZ-Tfs (0.6 mmol), FeTPPCl (3 mol%), NaOH (aq.)/DCM (5:1), 40 °C, 18 h

test the potential of this reagent in other carbenoid transformations. The Doyle–Kirmse reaction of allyl/propargyl thioethers was first studied, where we were delighted to find that reaction of TFHZ-Tfs with allylic thioethers catalyzed by Fe(TTP)Cl delivered the desired CF$_3$-substituted homoallyl or alkenyl products in excellent yields (Fig. 7)[46,47]. This reaction offers a direct and powerful method for the construction of C(sp$^3$)–S and C–C bonds by [2,3]-sigmatropic rearrangement of diazo-derived ylids[48,49], and again the use of TFHZ-Tfs proved superior for the generation of trifluorodiazoethane compared to the oxidation

of trifluoroethylamine[50]. The scope of this reaction was found to be quite broad, with aryl thioethers containing electron-donating and electron-withdrawing groups at different positions of aryl ring affording CF₃-substituted homoallyl products (**69–83**) in good to excellent yields. Thioethers bearing a disubstituted aryl group also proved suitable, such as a 2,5-dimethylated substituent, which gave product **84** in 84% yield. Naphthyl and heteroaryl allyl thioethers also proceeded efficiently to give the corresponding products **85** and **86** (80% and 75%, respectively). Alkyl allyl thioethers, including benzyl, methyl, and bisallyl substituents, were also well-tolerated to produce the desired products (**87–89**) in good yields.

**Scope of Doyle–Kirmse reaction with allyl thioether.** The effect of substituents (R¹, R², and R³) on the allyl thioether unit was similarly evaluated, which revealed that the substituent at the 2′-position substituent on the allyl thioether unit (R¹) could be varied (methyl, phenyl, or halogen), giving products **90–93** with high efficiency. Equally, a range of 1,2-disubstituted allylic thioethers (R² = Me/Ph, R³ = H/Me) were compatible, affording products (**94–98**) with high stereoselectivity (up to 10:1 *dr*, absolute configuration of major isomers was unambiguously confirmed by single crystal X-ray diffraction analysis of its derivatives, details see Supplementary Table 8). Most noteworthy among these variations is the rearrangement to generate a quaternary carbon center in homoallylic sulfide **96**. Moreover, a cyclic olefin-substituted thioether afforded the *S*-to-*C* transposition product **99** in 77% yield, with high stereoselectivity (7:1 *dr*).

**Scope of Doyle–Kirmse reaction with propargyl thioether.** We next investigated the scope of the Doyle–Kirmse reaction using propargyl thioethers. Pleasingly, aryl, alkyl, fused aryl, and heteroaryl-functionalized propargyl thioethers all reacted smoothly with TFHZ-Tfs to give the expected allene products in good to excellent yields (**100–103**). Internal alkynyl thioethers exhibited outstanding reactivity, as demonstrated by reactions of methyl-, phenyl- and TMS-substituted propargyl thioethers, which gave products **104–106** in 72–75% yield.

**Applications of TFHZ-Tfs in cyclopropanation.** Finally, the application of TFHZ-Tfs as a diazo precursor in cyclopropanation reactions was examined, with the aim of providing an alternative approach to medicinally-relevant trifluoromethylcyclopropanes[2,51–55]. To our delight, various terminal olefins underwent smooth reaction with TFHZ-Tfs under basic conditions in the presence of Fe(TPP)Cl, giving the desired CF₃-substituted cyclopropanes in high yields (Fig. 8). Good

functional group tolerance and excellent stereoselectivity (>20:1) were observed: electron-neutral, -rich, and -poor styrenes all underwent efficient cyclopropanations, affording the corresponding trifluoromethylcyclopropane products **107–113** in 81–95% yields. Again, naphthyl and heteroaryl groups were accommodated, providing products **114** and **115** in 87% and 81% yields respectively. Other conjugated dienes and enynes were examined, and also generated the corresponding cyclopropanes (**116–118**) in excellent yields. Finally, the use of 1,1-disubstituted olefins was equally well-tolerated in spite of increased steric hindrance, delivering trisubstituted products **119** and **120** without diminishing the reaction efficiency or stereoselectivity.

## Discussion

In summary, we report the development of trifluoroacetaldehyde N-tfsylhydrazone (TFHZ-Tfs)—a bench-stable crystalline reagent that represents a versatile trifluorodiazoethane surrogate, which can generate CF₃CHN₂ under basic conditions in a controlled manner that avoids excessive buildup of the hazardous diazo compound. A number of applications of TFHZ-Tfs are described, including the discovery of *gem*-difluoroalkenylation of X–H bonds (X = S, N, O, Se), Doyle–Kirmse rearrangements, and trifluoromethylcyclopropanation reactions, with superior performance over other sources of CF₃CHN₂. Considering the procedural advantages of this trifluorodiazoethane surrogate, and the importance of generally applicable fluorination methodologies, these findings create many opportunities for the wider exploration of the chemistry of trifluorodiazoethane.

## Methods

**General procedure for the synthesis of *gem*-difluorovinyl thioether.** A screw capped reaction vial was charged with TFHZ-Tfs (0.6 mmol), toluenethiol (0.3 mmol), Fe[P2] (0.003 mmol) and SDBS (sodium dodecylbenzenesulphonate) (0.09 mmol) under air, followed by addition of DCM (1.0 mL) and KOH *aq.* (5.0 mL, 20 wt%) (Fig. 5, Method A) . The resulting mixture was stirred at 40 °C for 18 h. Then 10 mL water was added to the mixture, which was extracted with DCM (3 × 10 mL). The organic layer was combined and dried with anhydrous MgSO₄, then filtered through a short silica gel eluting with DCM. The filtrate was evaporated under reduced pressure to leave a crude mixture, which was separated by flash column chromatography to afford the pure product.

**General procedure for the synthesis of *gem*-difluorovinyl amine.** A screw capped reaction vial was charged with TFHZ-Tfs (0.6 mmol), amine (0.3 mmol), Cu(OTf)₂ (0.06 mmol) and LiO*t*Bu (1.2 mmol), then evacuated and filled with argon for three times, followed by addition of DCE (3.0 mL) and toluene (1.0 mL) via syringe (Fig. 5, Method B). The resulting mixture was stirred at 40 °C for 24 h. The reaction crude was filtered through a short silica gel eluting with DCM. The filtrate was evaporated under reduced pressure to leave a crude mixture, which was separated by flash column chromatography to afford the pure product.

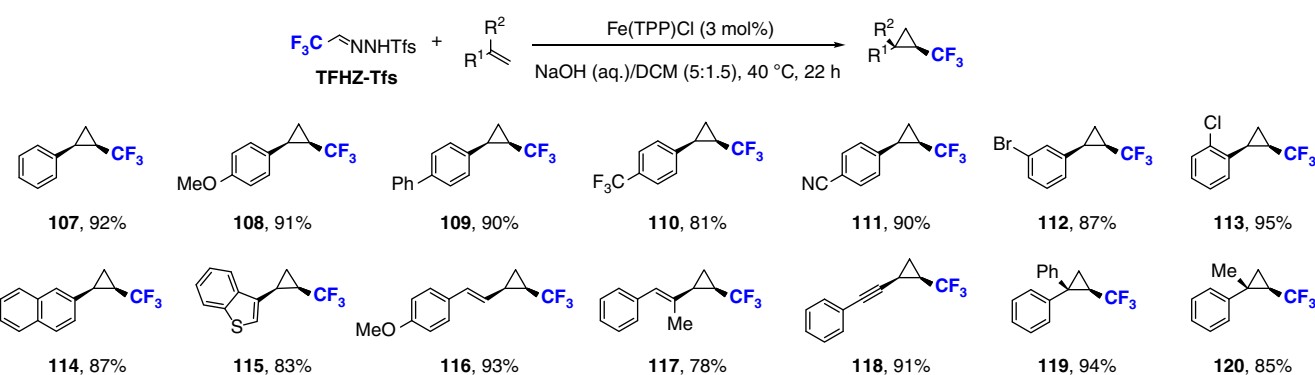

**Fig. 8** Scope of trifluoromethylcyclopropanation. Reaction conditions: olefin (0.3 mmol), TFHZ-Tfs (0.6 mmol), FeTPPCl (3 mol%), NaOH (aq.)/DCM (5:1.5), 40 °C, 22 h

**General procedure for the synthesis of *gem*-difluorovinyl ether**. A screw capped reaction vial was charged with TFHZ-Tfs (1.0 mmol), and NaH (60 wt%, 2 mmol) and was evacuated and filled with argon for three times, followed by addition of dry DCE (8.0 mL) via syringe. The resulting mixture was stirred at room temperature for 1 h (Fig. 5, Method C). Then, alcohol (0.5 mmol) and CuBr (0.15 mmol) were added and the system was stirred at 40 °C for 24 h. The reaction crude was filtered through a short silica gel eluting with DCM. The filtrate was evaporated under reduced pressure to leave a crude mixture, which was purified by column chromatography on silica gel.

**General procedures for Doyle–Kirmse reaction**. A screw capped reaction vial was charged with TFHZ-Tfs (0.6 mmol), FeTPPCl (0.009 mmol), then evacuated and filled with argon for three times, then DCM (1 mL) which dissolved with allyl or propargyl sulfide (0.3 mmol) and NaOH aq. (5 mL, 20 wt%) was successively added by syringe (Fig. 7). The reaction was stirred at 40 °C for 18 h. Then 10 mL water was added to the mixture and layers partitioned. The aqueous layer was extracted with DCM (3 × 10 mL) and the organic layer was combined and dried with anhydrous MgSO$_4$, then filtered through a short silica gel eluting with DCM. The filtrate was evaporated under reduced pressure to leave a crude mixture, which was purified through silica gel flash column chromatography eluting with *n*-hexane to give the final product.

**General procedures for cyclopropanation reaction**. A screw capped reaction vial was charged with TFHZ-Tfs (0.6 mmol), FeTPPCl (0.009 mmol), then evacuated and filled with argon for three times, then DCM (1.5 mL) which dissolved with styrene (0.3 mmol) and NaOH aq. (5 mL, 20 wt%) was successively added by syringe (Fig. 8). The reaction was stirred at 40 °C for 22 h. Then 10 mL water was added to the mixture and layers partitioned. The aqueous layer was extracted with DCM (3 × 10 mL) and the organic layer was combined and dried with anhydrous MgSO$_4$, then filtered through a short silica gel eluting with DCM. The filtrate was evaporated under reduced pressure to leave a crude mixture, which was purified through silica gel flash column chromatography eluting with *n*-hexane to give the final cyclopropane product.

## Data availability

The authors declare that all the data supporting the findings of this study are available within the paper and its supplementary information files, or from the corresponding author upon request. The X-ray crystallographic coordinates for structures reported in this article have been deposited at the Cambridge Crystallographic Data Center (Trifluoroacetaldehyde N-tfsylhydrazone: CCDC 1814685, Trifluoroacetaldehyde N-tosylhydrazone: CCDC 1814683, Trifluoroacetaldehyde N-nosylhydrazone: CCDC 1827227, 9′: CCDC 1814506, 95′: CCDC 1881268). These data could be obtained free of charge from The Cambridge Crystallographic Data Center via https://www.ccdc.cam.ac.uk/structures/.

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

## Acknowledgements

Financial support by NSFC (21871043, 21522202, 21502017) and Department of Science and Technology of Jilin Province (20180101185JC, 20190701012GH). EAA thanks the EPSRC for support (EP/M019195/1).

## Author contributions

X.Z. and Z.L. contributed equally to this work. X.Z., Z.L., X.Y., and Y.D. performed the experiments. M.V. and G.Z. conducted mechanistic studies. E.A.A. participated in the discussion of the research results as well as the revision of the manuscript. X.Z. and X.B. wrote the manuscript.

## Additional information

**Competing interests:** The authors declare no competing interests.

