## [Peer Review File · Nature Communications]

Reviewers' comments:

Reviewer #1 (Remarks to the Author):

This manuscript describes a new trifluorodiazaoethane surrogate in the form of Trifluoroacetaldehyde N-tfsylhydrazone, which the authors have made using simple condensation chemistry. The reagent is demonstrated to effect the difluoroalkenylation of amines, thiols, selenols and alcohols.

The authors correctly state in the introduction that there are hazards associated with trifluorodiazaoethane and that difluoroalkenes are important structural motifs in medicinal chemistry. I don't think the authors can argue (and they have not tried to do so) that the heteroatom-substituted fluoralkene products are of value at present but this is because they are currently very hard to obtain by other methods. I think that medicinal chemists would be very interested in exploring this heteroatom modification process. I think the authors should add two citations to the introduction:

Tetrahedron 1987, 43, 4390 gives an Se-based reagent for making the S-fluoroalkenes

Nat. Commun. 2017, 8, 15913 describes a method to access trifluoroethylated secondary amines using trifluoroacetic acid (Figure 1a gives the impression that the only way to do this is by using trifluorodiazaoethane).

I think that the authors have done a good job in demonstrating the wide utility of this method and very many novel fluorine-containing compounds are described which will be of particular interest to medicinal chemists. In terms of practicality some of the Fe-porphyrins require careful conditions/glove box chemistry to make which imposes a barrier in terms of others using the chemistry but, overall, I think that this is justified by the wide scope and the fact that this work opens up this interesting class of difluoroalkenes with heteroatom substituents. In summary, I think that this manuscript could be suitable for publication but there are some concerns that need to be addressed first. My recommendation is to give the authors the opportunity to make these additions and corrections (listed below) before a final decision is made.

Points to address main text:

Scheme 2 – some very interesting further transformations are shown here. The main question I have is can the double bond in any of the products be hydrogenated to give the difluoroethylated compounds? There is a great deal of interest in difluorinated amines and ethers currently and this simple transformation would give rise to another class of very valuable products. I am not suggesting that this be done for lots of examples but, if possible, add one example of this to Scheme 2. If it doesn't work then a short statement about the conditions attempted would be useful because I think readers will be looking to see if this can be done.

Are Boc protected amines tolerated in any of the reactions? Again one example would be nice to see as substrates used in medicinal chemistry will often be nitrogen rich and synthesis chemists in industry would want to know if NBoc amines are tolerated.

Does the reaction work with aliphatic amines? If not what happens? There is a statement about primary amines in the text but this looks like it means primary anilines. Can for example butylamine be used?

In the experimental section there are several points that must be addressed:

All novel compounds must have full characterisation data. There is no HRMS data for any of the products. This is required to corroborate the empirical formulae of novel compounds unless the authors have elemental analysis data. There is no IR data – again this is absolutely required for novel compounds.

¹³C chemical shifts are reported to 1 decimal point by convention and not 2.

There are some errors in ¹H NMR analysis of multiplicities – quite a few of these but one example is given but all need to be corrected. Compound 112 it not possible to have any aromatic triplets here because the Hs are chemically inequivalent. Authors should be using dd here or apparent triplet.

I am not sure how the NOESY has been interpreted and how this corroborates the relative stereochemistry of 95. Authors need to give a much more lucid explanation with conformational drawings to support this. Alternatively, derivatize the alkene and get an X-ray structure.

The significant figures of the masses and molar amounts of reagents should be consistent within every set of parentheses. For example, (6 mg, 0.003 mmol) (line 17) is not correct because the mmol quantity is expressed with more accuracy than the mg quantity. Needs correcting throughout.

Reviewer #2 (Remarks to the Author):

Fluoro-compounds are of particular interest in the fields of materials, agrochemicals, and pharmaceuticals due to the unique characteristics of the Fluoro-atom such as high electronegativity, electron density, and hydrophobicity. Hence, it has been of great synthetic interest to develop an efficient methods for incorporation of F into organic molecules. However, this is highly depended of the availability of fluorinating reagents, which are a class of special reagents and often hazardous.

In this manuscript, Prof. Bi et al reported the development of trifluoroacetaldehyde N-tfsylhydrazone (TFHZ-Tfs) as a new and safer trifluorodiazaoethane surrogate. They first describe the large-scale preparation and safety evaluation of TFHZ-Tfs. Next, they discovered that the gem-difluoroalkenylation of p-methylthiophenol with the reagent TFHZ-Tfs, delivered in situ trifluorodiazaoethane under basic conditions, could be catalyzed by Fe[P2] (1% mol). They thus developed a new method for the synthesis of gem-difluoroalkenylation of X-H (X= N, O, S, Se). Next, the use of the products (heteroatom-substituted gem-difluoroalkenes), in turn, as building blocks was examined. In addition, applications of this new reagent to the Doyle-Kirmse rearrangement and trifluoromethylcyclopropanation reactions have also been explored, which demonstrated superior outcomes to approaches using pre-formed CF₃CHN₂. Given the importance of this generally applicable fluorination methodology, the use of TFHZ-Tfs would provide novel opportunities for both organic and medicinal chemistry as well as related field. On the other hand, the manuscript is well written, and the SI has been prepared with care. As such this manuscript may be acceptable for a publication in Nature Communications subjecting to some minor alternations noted below.

1. As a key element, the preparation, availability and application of FeTPPCL and Fe[P1,2] should be commented and references cited.
2. Page 1. "We targeted the use of N-sulfonylhydrazones as a new trifluorodiazaoethane surrogate" changes to "We targeted the use of trifluorinated N-sulfonylhydrazones as a new trifluorodiazaoethane surrogate. "
3. Please cite this reference: Angew. Chem. Int. Ed. 2014, 53, 11575 –11578.
4. It lacks IR data for most compounds and some MS data.

Reviewer 1:

This manuscript describes a new trifluorodiazaoethane surrogate in the form of Trifluoroacetaldehyde N-tf-sylhydrazone, which the authors have made using simple condensation chemistry. The reagent is demonstrated to effect the difluoroalkenylation of amines, thiols, selenols and alcohols.

The authors correctly state in the introduction that there are hazards associated with trifluorodiazaoethane and that difluoroalkenes are important structural motifs in medicinal chemistry. I don't think the authors can argue (and they have not tried to do so) that the heteroatom-substituted fluoralkene products are of value at present but this is because they are currently very hard to obtain by other methods. I think that medicinal chemists would be very interested in exploring this heteroatom modification process. I think the authors should add two citations to the introduction: *Tetrahedron* 1987, 43, 4390 gives an Se-based reagent for making the S-fluoroalkenes *Nat. Commun.* 2017, 8, 15913 describes a method to access trifluoroethylated secondary amines using trifluoroacetic acid (Figure 1a gives the impression that the only way to do this is by using trifluorodiazaoethane).

According to the reviewer's suggestions, the references have been added.

I think that the authors have done a good job in demonstrating the wide utility of this method and very many novel fluorine-containing compounds are described which will be of particular interest to medicinal chemists. In terms of practicality some of the Fe-porphyrins require careful conditions/glove box chemistry to make which imposes a barrier in terms of others using the chemistry but, overall, I think that this is justified by the wide scope and the fact that this work opens up this interesting class of difluoroalkenes with heteroatom substituents. In summary, I think that this manuscript could be suitable for publication but there are some concerns that need to be addressed first. My recommendation is to give the authors the opportunity to make these additions and corrections (listed below) before a final decision is made.

Points to address main text:

Scheme 2 – some very interesting further transformations are shown here. The main question I have is can the double bond in any of the products be hydrogenated to give the difluoroethylated compounds? There is a great deal of interest in difluorinated amines and ethers currently and this simple transformation would give rise to another class of very valuable products. I am not suggesting that this be done for lots of examples but, if possible, add one example of this to Scheme 2. If it doesn't work then a short statement about the conditions attempted would be useful because I think readers will be looking to see if this can be done.

Are Boc protected amines tolerated in any of the reactions? Again one example would be nice to see as substrates used in medicinal chemistry will often be nitrogen rich and synthesis chemists in industry would want to know if NBoc amines are tolerated.

We tried Boc protected amines, but they did not react.

Does the reaction work with aliphatic amines? If not what happens? There is a statement about primary amines in the text but this looks like it means primary anilines. Can for example butylamine be used?

It should be anilines rather than amines, which we have changed in the text. We have also tried butylamine and other aliphatic amines, but did not get the desired product, probably due to the fact that the electron-withdrawing ability of the two fluorine atoms is not enough to stabilize the aliphatic enamine (Normally, an aliphatic enamine requires strong electron withdrawing group to stabilize, such as cyano, ester or carbonyl, etc.).

All novel compounds must have full characterisation data. There is no HRMS data for any of the products. This is required to corroborate the empirical formulae of novel compounds unless the authors have elemental analysis data. There is no IR data – again this is absolutely required for novel compounds.

According to the reviewer's comments, we supplemented nearly all most IR, MS, and HRMS data.

^{13}C chemical shifts are reported to 1 decimal point by convention and not 2.

We have corrected this in the supporting information

There are some errors in ^1H NMR analysis of multiplicities – quite a few of these but one example is given but all need to be corrected. Compound 112 it not possible to have any aromatic triplets here because the Hs are chemically inequivalent. Authors should be using dd here or apparent triplet.

According to the reviewer's opinion, we re-examined the spectrum. Although Hs is not equivalent in some compounds, the chemical shift of Hs is very close (even overlapping) due to the similar chemical environment. In this case, we have enlarged some of the spectra to give the reader a more accurate view of this phenomenon.

I am not sure how the NOESY has been interpreted and how this corroborates the relative stereochemistry of 95. Authors need to give a much more lucid explanation with conformational drawings to support this. Alternatively, derivatize the alkene and get an X-ray structure.

In the supporting information we supplied the conformational drawings and marked the chemical shifts of the main hydrogen. Above the NOESY spectrum we give the details of the analysis so that the reader can clearly understand the spatial correlation between Hs.

The significant figures of the masses and molar amounts of reagents should be consistent within every set of parentheses. For example, (6 mg, 0.003 mmol) (line 17) is not correct because the mmol quantity is expressed with more accuracy than the mg quantity. Needs correcting throughout.

According to the reviewer's opinion, we have corrected the significant figures.

Reviewer2:

Fluoro-compounds are of particular interest in the fields of materials, agrochemicals, and pharmaceuticals due to the unique characteristics of the Fluoro-atom such as high electronegativity, electron density, and hydrophobicity. Hence, it has been of great synthetic interest to develop an efficient methods for incorporation of F into organic molecules. However, this is highly depended of the availability of fluorinating reagents, which are a class of special reagents and often hazardous.

In this manuscript, Prof. Bi et al reported the development of trifluoroacetaldehyde N-tfsylhydrazone (TFHZ-Tfs) as a new and safer trifluorodiazethane surrogate. They first describe the large-scale preparation and safety evaluation of TFHZ-Tfs. Next, they discovered that the gem-difluoroalkenylation of p-methylthiophenol with the reagent TFHZ-Tfs, delivered in situ

trifluorodiazaoethane under basic conditions, could be catalyzed by Fe[P2] (1% mol). They thus developed a new method for the synthesis of gem-difluoroalkenylation of X-H (X= N, O, S, Se). Next, the use of the products (heteroatom-substituted gem-difluoroalkenes), in turn, as building blocks was examined. In addition, applications of this new reagent to the Doyle-Kirmse rearrangement and trifluoromethylcyclopropanation reactions have also been explored, which demonstrated superior outcomes to approaches using pre-formed CF₃CHN₂. Given the importance of this generally applicable fluorination methodology, the use of TFHZ-Tfs would provide novel opportunities for both organic and medicinal chemistry as well as related field. On the other hand, the manuscript is well written, and the SI has been prepared with care. As such this manuscript may be acceptable for a publication in Nature Communications subjecting to some minor alternations noted below.

1. As a key element, the preparation, availability and application of FeTPPCl and Fe[P1,2] should be commented and references cited.

We added a discussion of iron porphyrins in the paper and cited the literature.

2. Page 1. “We targeted the use of N-sulfonylhydrazones as a new trifluorodiazaoethane surrogate” changes to “We targeted the use of trifluorinated N-sulfonylhydrazones as a new trifluorodiazaoethane surrogate.”

According to the reviewer's comments, we have changed this in the text.

3. Please cite this reference: Angew. Chem. Int. Ed. 2014, 53, 11575 –11578.

According to this, we have cited this reference.

4. It lacks IR data for most compounds and some MS data.

According to this, we supplemented all most IR data, MS data and HRMS data.

Reviewers' comments:

Reviewer #1 (Remarks to the Author):

The authors have not addressed all of my points related to the SI in this revised manuscript. Therefore, I am afraid that it remains unsuitable for publication in this or any other journal. All journals require full characterisation for novel compounds.

The significant figures of the masses and molar amounts of reagents should be consistent within every set of parentheses. For example, (6 mg, 0.003 mmol) (line 17) is not correct because the mmol quantity is expressed with more accuracy than the mg quantity. Needs correcting throughout.

Although minor this has not been corrected throughout.

According to the reviewer's comments, we supplemented nearly all most IR, MS, and HRMS data.

This is not a minor point. IR and HRMS (NOT MS) data are absolute requirements for all novel compounds. The purpose of HRMS is to establish the empirical formula. This used to be done by elemental analysis but is now most commonly done through HRMS. This information is not gained from MS.

A conformational diagram has been added but the authors have simply drawn a single conformation for the diastereoisomer that they believe to be present! This is not what I asked for.

Having looked at this again I could draw or model the alternative diastereomer and, owing to free rotations, there are conformers in which the NOEs would be present. The authors need to appreciate the fact that there are very many conformers and the NMR experiments show the Boltzman-weighted average. An X-ray structure of a derivative of the sulfide would be much better and allow assignment of the relative stereochemistry with confidence.

It would be a embarrassing for the authors if the stereochemical assignment was later proved to be incorrect.

We have performed additional experiments and made revisions to the manuscript according to the comments of the reviewers. Below is our response to the reviewers point by point:

Reviewer 1:

Reviewers' comments:

The authors have not addressed all of my points related to the SI in this revised manuscript. Therefore, I am afraid that it remains unsuitable for publication in this or any other journal. All journals require full characterisation for novel compounds. The significant figures of the masses and molar amounts of reagents should be consistent within every set of parentheses. For example, (6 mg, 0.003 mmol) (line 17) is not correct because the mmol quantity is expressed with more accuracy than the mg quantity. Needs correcting throughout. Although minor this has not been corrected throughout.

According to the suggestion, we have re-examined and corrected all significant figures in the SI.

According to the reviewer's comments, we supplemented nearly all most IR, MS, and HRMS data. This is not a minor point. IR and HRMS (NOT MS) data are absolute requirements for all novel compounds. The purpose of HRMS is to establish the empirical formula. This used to be done by elemental analysis but is now most commonly done through HRMS. This information is not gained from MS.

According to the reviewer's comments, we have supplemented the full characterization data of all new compounds, including ^1H , ^{13}C , ^{19}F , IR and HRMS.

A conformational diagram has been added but the authors have simply drawn a single conformation for the diastereoisomer that they believe to be present! This is not what I asked for. Having looked at this again I could draw or model the alternative diastereomer and, owing to free rotations, there are conformers in which the NOEs would be present. The authors need to appreciate the fact that there are very many conformers and the NMR experiments show the Boltzman-weighted average. An X-ray structure of a derivative of the sulfide would be much better and allow assignment of the relative stereochemistry with confidence. It would be a embarrassing for the authors if the stereochemical assignment was later proved to be incorrect.

According to the reviewer's comments, we have oxidized compound **95** to a sulfone derivative **95'**. Absolute configuration of compound **95'** was unambiguously confirmed by single crystal X-ray diffraction analysis. This evidence has been added to the Supporting Information Section VI.

CCDC No. 1881268

REVIEWERS' COMMENTS:

Reviewer #1 (Remarks to the Author):

Having examined the manuscript for a third time I am now satisfied that the experimental section is at publication standard.

The significant figures are still not correct in many places but, as I said, this is a minor issues and the authors are obviously content for it to appear in this form.

My main concerns were IR and HRMS data which are now present and the stereochemical issue which is now resolved by doing the oxidation as suggested.

I can now recommend publication but I would urge the authors in future to take proper care in the preparation of the supporting information. Low quality SI will not be accepted and reflects poorly on the level of scholarship of the principal investigator who is ultimately responsible.